# Vertigo Associated with Otosclerosis and Stapes Surgery—A Narrative Review

**DOI:** 10.3390/medicina59081485

**Published:** 2023-08-18

**Authors:** Violeta Necula, Alma Aurelia Maniu, László-Péter Ujváry, Maximilian-George Dindelegan, Mara Tănase, Mihai Tănase, Cristina Maria Blebea

**Affiliations:** Otorhinolaryngology Department, “Iuliu Haţieganu” University of Medicine and Pharmacy, 400347 Cluj-Napoca, Romania

**Keywords:** otosclerosis, vertigo, dizziness, stapes surgery

## Abstract

Otosclerosis is a pathological condition affecting the temporal bone, and is characterized by remodelling of the labyrinthine bone tissue through a dynamic process of osteolysis and osteogenesis. This condition progressively leads to hearing loss, tinnitus, and vertigo. Stapedotomy, a surgical procedure involving the removal of the stapes superstructure and its replacement with a prosthesis, is the treatment of choice to improve hearing in individuals with otosclerosis. However, vestibular dysfunction is a significant complication associated with this procedure, which can occur intraoperatively or postoperatively, ranging from the immediate postoperative period to weeks, months, or even years after surgery. This paper aims to provide a comprehensive review of the most important causes of vertigo associated with otosclerosis and stapes surgery with the goal of minimizing the incidence of this complication. Understanding the underlying factors contributing to vertigo in this context is crucial for the prevention and effective management of vertigo in patients undergoing stapedotomy.

## 1. Introduction

Otosclerosis, also known as otospongiosis, is a progressive primary bone disorder of the otic capsule. It is characterized by abnormal focal resorption and recalcification of the endochondral layer of the temporal bone. This results in progressive conductive hearing loss, evolving in severe cases to a combination of conductive and sensorineural hearing loss. The otosclerotic process starts in the anterior part of the oval window, near the fissula ante fenestram resulting in fenestral otosclerosis. The disease can progress beyond the fissula ante fenestram and extend to the pericochlear otic capsule in cochlear otosclerosis or retrofenestral otosclerosis. The advanced stages can affect structures such as round window, semicircular canals, labyrinth, or vestibular nerve endings [1].

While progressive hearing loss is the main complaint in otosclerosis, other symptoms such as tinnitus and balance disorders are frequently associated with the condition. Up to 30% of patients with otosclerosis may experience vestibular symptoms, including instability, dizziness, and vertigo [2]. These symptoms can occur before or after treatment, and may arise as immediate or delayed complications of stapes surgery.

Surgical intervention is the preferred treatment for otosclerosis, with the aim of restoring the mechanism of sound transmission from the ossicular chain to the inner ear receptors, resulting in an air–bone gap closure of less than 10 dB in over 80% of cases [3,4]. Stapedotomy has become the most commonly performed surgical technique, replacing stapedectomy. For patients who are not suitable for or decline surgery, hearing aids can be recommended to improve hearing [5].

Considering the significant impact of vertigo on quality of life, this study aims to evaluate the presence of vertigo in otosclerosis and assess the risk of developing vestibular disorders following stapes surgery. By understanding the relationship between otosclerosis, vertigo, and stapes surgery, clinicians can better manage and minimize the incidence of this distressing complication.

## 2. The Otosclerotic Process

The otosclerotic process consists of abnormal replacement of enchondral bone with cancellous bone and subsequently with sclerotic bone. The process occurs in waves of osteolysis followed by osteogenesis. During the active phase of otospongiosis, the resorption process consists of replacing the normal bone around the blood vessels, which has cellular fibrous connective tissue, with mononuclear histiocytes, osteocytes, and osteoclasts [6]. A number of studies have shown that the enzymes secreted by the cells from the otospongiotic foci play a role in bone decalcification [7,8], while other studies have investigated the level of alkaline phosphates in the decalcification process of the otic capsule [9]. The final stage of the process is the otosclerosis stage, when the bone becomes mineralized and presents a mosaic appearance [10].

Lim et al. [11] described three types of otosclerotic lesions: cellular or spongiotic, characterized by the activation of monocytes, macrophages, osteoblasts, and osteoclasts; fibrotic, in which extensive bone fibrosis occurs; and sclerotic, characterized by a marked reduction of bone cells. Chevance et al. [12] reported the presence of osteolytic enzymes in the perilymph of patients who underwent surgery for otosclerosis, and suggested that these enzymes may have an important role in the development of otosclerotic lesions in the inner ear.

Stapes fixation occurs due to calcification of the annular ligament and the invasion of otosclerotic lesions at the oval window [13]. Otosclerotic lesions in the cochlear endosteum can lead to atrophic and hyalinization changes of the spiral ligament [14]. Damage to the spiral ligament can disrupt the chemical balance of ion-fluid recycling [15] and obstruct the endolymphatic duct and sac, resulting in biochemical changes [16,17]. Gros et al. [18] observed that vestibular disorders are frequently associated with sclerotic lesions. Saim and Nadol [19] reported that patients with vestibular symptoms have elevated bone-conduction thresholds and suggested that the degeneration of the vestibular nerve and Scarpa ganglion cells could be responsible for these symptoms, regardless of otosclerotic damage to the vestibular end organs.

Otosclerosis is associated with inflammation, disturbed collagen expression, and the presence of viral receptors and antigens in the otosclerotic foci [20]. At the molecular level, the bone remodelling process is regulated by a series of cytokines, signalling molecules that play a crucial role in regulating various cellular processes including bone remodelling. In otosclerosis, cytokines such as osteoprotegerin (OPG), receptor activator of nuclear factor kB (RANK), and RANK liand (RANKL), as well as transforming growth factor ß1 (TGF-ß1), are involved in controlling the balance between bone resorption and bone formation [20,21]. The presence of the measles virus and concurrent inflammation may trigger the abnormal bone remodeling that is a characteristic of otosclerosis. In the active phase, there is an increase in inflammation, detectable measles virus particle, local expression of tumor necrosis factor alpha (TNF-α), and negativity for OPG expression. During this phase, the balance between bone resorption and formation may be disrupted. In the inactive phase of otosclerosis, OPG positivity and TNF-α negativity are observed, along with absence of inflammation [22,23]. The increased level of TNF-α can stimulate osteoclast activation, induce RANKL expression, and reduce osteoclast apoptosis. This sequence of events ultimately leads to osteolysis and contributes to the process of otospongiosis [22,24]. TNF-α overexpression stimulates osteoclast formation both by inhibiting OPG secretion and by stimulating RANKL formation [23,24].

## 3. Anatomy of the Membranous Labyrinth

The anatomy of the membranous labyrinth is essential to understanding certain pathological processes; in surgical procedures involving the inner ear, these particularly relate to otosclerosis and stapes surgery. The middle ear communicates with the inner ear through the oval and round windows. The footplate articulates with the oval window through the annular ligament. Beyond the oval window is the vestibule, which is filled with perilymph. The membranous labyrinth is supported by periotic connective tissue within the perilymphatic space, which is medial and superior to the utricle and saccule and absent lateral to them [25].

The otolith organs of the vestibule are the macula of the utricle and saccule, located medially in the vestibule. Their role is to detect the position and direction of the head as well as the linear and gravitational acceleration [26].

The utricle has an elongated shape and communicates with the semicircular canals. On the inferior wall, the more lateral is the macula of the utricle, oriented horizontally. It is localized next to the upper edge of the oval window at 0.5–1 mm distance [27]. In surgical procedures, the macula of the utricle can be observed as a white plaque within the vestibule [28].

The saccule is situated in the spherical recess, and its macula has a vertical orientation perpendicular to the macula of the utricle. The anterior wall of the saccule is adjacent to the footplate. Between these two structures is found a connective tissue, named the reinforced area of the saccular membrane [29]. The macula of the saccule projects below the horizontal line passing through the arm of the stapes [30]. The distance between the saccule and the anterior edge of the oval window is between 1 and 1.5 mm [27]. The saccule communicates with the utricle through the utriculo-saccular duct, with the cochlea through the reuniens duct, and with the endolymphatic duct through the sinus of the duct.

The membrana limitans is a membranous structure, similar to a network, which delimits the superior vestibular labyrinth from the inferior part, laying below the utricle and supporting its macula [31]. In certain cases, the membrana limitans can present thin fibrillary attachments to the footplate, especially in the posterior third [25]. Its role is more of a support than a barrier, as it has a discontinuous structure that allows the passage of perilymph. Its insertion is in the superior part of the vestibule, superior to the oval window, immediately above the stapes footplate. In certain cases, the membrana limitans can be directly inserted in the footplate [25]. Pauw et al. [32] reported that the distance between the footplate and the utricle is smaller in patients with otosclerosis than in normal subjects. These rapports and adhesions of the labyrinthine structures to the footplate may partially explain the vertigo experienced during stapes surgery.

Vestibular symptoms can be part of the clinical manifestations of otosclerosis or can occur during or after stapes surgery, either immediately or with delayed onset.

## 4. Preoperative Vertigo

The incidence of preoperative vestibular symptoms in otosclerosis patients varies greatly from one study to another. Different studies have reported incidence ranging from 8.6 to 30% [33,34,35]. While the exact cause of vertigo in patients with otosclerosis is not fully understood, several factors have been proposed (Table 1).

One factor is the presence of otosclerotic foci, which can affect the endolymphatic duct and sac, leading to hydrops [15]. This abnormal accumulation of fluid in the inner ear can contribute to vertigo. Temporal bone studies have shown the presence of endolymphatic hydrops (EH) in specimens with extensive otosclerotic lesions in the cochlear endosteum or in the vestibular aqueduct, obstructing the flow of endolymph and disrupting labyrinthine fluid homeostasis [15,16]. The presence of endolimphatic hydrops can be visualized on delayed three-dimensional (3D) fluid-attenuated inversion recovery (FLAIR) MRI images obtained after intravenous administration of gadolinium. Sone et al. [36] suggested that the presence of preoperative asymptomatic vestibular EH could serve as a predictive factor for postoperative complications following stapes surgery. EH located adjacent to the oval window could be a contraindication for stapes surgery. The proximity of EH to the surgical site may increase the risk of vestibular complications after the procedure, including vertigo.

Detachment of the otoconia from the macula of the utricle is another factor that could contribute to positional vertigo, and may explain anomalies observed in the ocular and cervical vestibular-evoked myogenic potentials (oVEMP and cVEMP), which are used to assess the status of the utricle and saccule, respectively [37,38,39].

The third factor is vestibular end organ and/or neural degeneration due to otosclerotic foci involving the utricular or ampullary nerve. These changes could be related to the utricular deficit and oVEMP anomalies in patients with vertigo and otosclerosis [40].

Additionally, hydrolytic enzymes originating in the otosclerotic foci have been identified in the perilymph of otosclerosis patients; these can produce vascular and neuroepithelial lesions. Moreover, the cytokines produced in these foci can cause changes in labyrinthine fluids’ chemical composition and homeostasis [41].

Degeneration of receptor cells in the vestibule and changes in the nonsensory epithelium have been observed in temporal bone studies of otosclerosis patients. Kaya et al. [42] studied temporal bones harvested from patients diagnosed with otosclerosis and found a decrease in the population of vestibular dark cells and vestibular transitional cells in temporal bone specimens with endosteal involvement. The role of these cells is to maintain the homeostasis of the labyrinthine fluid by controlling the transport of ions and water in order to prevent vestibular dysfunction. Another study by Hizli et al. [43] found that the mean density of type I hair cells in the saccule was significantly reduced in cases with endosteal involvement, suggesting that the extension of the otosclerotic foci towards the endosteum may be an important factor in the occurrence of vestibular symptoms in patients with otosclerosis. They suggested that this might explain the abnormal oVEMP and cVEMP response in patients with otosclerosis and vestibular symptoms.

Saka et al. [44] studied the vestibular-evoked myogenic potential in response to bone-conducted sound (BC-VEMP) in a group of 25 patients and showed that 9 of 10 patients with vestibular dysfunction presented abnormal BC-VEMPs. This suggests that saccular dysfunction may be involved in these patients, possibly due to saccular hydrops or the extension of otosclerotic foci to the saccular macula or saccular afferent, considering the anatomical proximity of the saccule to the oval window.

On the other hand, another study involving 27 patients with otosclerosis and vertigo reported abnormal oVEMPs in response to impulsive stimulation, suggesting pathological abnormalities related to the utricle [40]. Hayasi et al. [37] studied 35 temporal bones with otosclerosis, reporting a higher incidence of cupular deposits compared to temporal bone without otosclerosis. They suggested that the origin of these deposits was probably the otoconia from the utricle, from where they detached and migrated to the cupula of semicircular canals.

The specific manifestation of vestibular symptoms can vary from person to person, and may be influenced by factors such as the extent of otosclerotic involvement and individual differences in anatomy and physiology. A study conducted by Eza-Nuñez et al. [45] highlighted the diversity of vestibular symptoms experienced by patients with otosclerosis. Patients with otosclerosis mention vertigo or imbalance, which can manifest in different ways, including a single episode or recurrent attacks either triggered by positional changes or occurring spontaneously. In their study, positional vertigo was associated with otosclerosis in 32.5% of patients and Ménière syndrome was reported in 30% of patients. Around 27.5% of patients experienced spontaneous recurrent vertigo, approximately 7.5% of patients presented with chronic unrelapsing imbalance, and a small percentage of patients (2.5%) had acute unilateral vestibulopathy [45].

These findings suggest that multiple factors, including endolymphatic hydrops, detachment of otoconia, degeneration of receptor cells, and cupular deposits, may lead to vertigo in otosclerosis patients. Further research is needed in order to fully understand the underlying mechanisms and develop targeted interventions for vestibular symptoms in this population.

## 5. Immediate Postoperative Vertigo

Treatment of otosclerosis is mainly surgical, generally with good results. The most common technique is stapedotomy, a minimally invasive technique that has largely replaced stapedectomy due to having fewer complications, including vestibular disorder [33]. The surgical approach can be either classical, using a microscope, or endoscopic, and the stapedotomy can be carried out by a conventional or laser-assisted technique.

Vertigo is reported to occur intraoperatively in 2.1% of patients, mainly due to the manipulation of the footplate. It may occur due to frequent suctioning in the middle ear, and less often to a floating footplate. This is treated by reassurance of the patient and vertiginous drugs [46] (Table 1).

After surgical treatment, immediate postoperative vertigo can occur in a significant percentage of patients. The reported incidence of vestibular symptoms varies among studies, ranging from 3.4% to 70% [47,48,49] (Table 1).

Early postoperative vertigo is usually temporary, and authors report remission of symptoms in most cases after 5 to 7 days with conservative management, including medication and bed rest [50,51].

The use of a CO_2_ Laser in footplate perforation has been suggested to reduce the prevalence of postoperative vertigo due to minimal mechanical trauma to the inner ear through lesser footplate manipulation [47,52].

Several factors contribute to the occurrence of immediate postoperative vertigo. A possible cause of premature vertigo could be serous and chemical labyrinthitis, which involves irritation of the membranous labyrinth, particularly the macula of the utricle located near the oval window [53]. Suction of the perilymph from the vestibule or contact of the instrument with the membranous labyrinth can trigger vertigo [54]. According to measurements taken Pauw et al., penetration of instruments or the prosthesis into the vestibule is considered less risky in the centre and lower third of the oval window [32].

Nystagmus is observed postoperatively in approximately 65.7% of patients, and may persist for over one month, as shown by Fukuda et al. [55] in a study conducted in 2021.

Singh et al. [52] used posturography to evaluate patients, and found that patients experienced vestibular deficits and increased subjective symptom scores at one week after surgery, with remission occurring after four weeks.

Assessment of cVEMP with air conducted stimuli before surgery and three months after stapedotomy showed a significant reduction in the amplitude of P1/N2 waves in patients who complained of dizziness and vertigo, suggesting a saccular lesion in these patients [56]. The reduction of air conduction (AC) and bone conduction (BC) VEMPs in patients with otosclerosis was reported by Trivelli et al. [57], with the observation that although the air conduction thresholds improved after surgery in all patients, AC-VEMP and BC-VEMP did not significantly improve in operated patients. Akazawa et al. [58] evaluated the cervical and ocular VEMPs through bone-conducted vibration before and after surgery, finding no significant changes in VEMPs in the operated ear after stapes surgery.

Postoperative vertigo following stapedotomy may be attributed to traumatization of the utricle, release of proteolytic enzymes, antigen–antibody reactions, pressure changes in labyrinthine fluids, and reduction of blood supply to the labyrinth caused by a floating footplate during the operation [53,59].

Among the three semicircular canals (SCC), the lateral SCC appears to be the most affected in both otosclerosis and after stapes surgery. SCC function can be evaluated by vHIT. Postoperative vHIT results have indicated subclinical damage to the lateral and posterior SCC. This is further supported by studies on temporal bones which revealed degeneration of the sensory epithelium in the cristae of the SCC [60,61]. Kujala et al. [62] evaluated patients after stapes surgery and found latent spontaneus horizontal-torsional nystagmus in 33% of patients on the day of surgery. The presence of this nystagmus suggests minimal impairment of the SCC.

Overall, immediate postoperative vertigo is a common occurrence following surgery for otosclerosis, though it is usually temporary and resolves with conservative management. Monitoring vestibular function through VEMPs and other tests can provide insights into the underlaying mechanisms and help to evaluate the impact of surgery on vestibular function.

## 6. Late Postoperative Vertigo

Late postoperative vertigo can occur following stapes surgery for otosclerosis, with a reported incidence ranging from 0.5% to 17% [54]. The persistence of vertigo beyond four weeks is observed in approximately 4% of patients, as shown by Birch et al. [63] in a study of 722 patients, while in Albera’s study 17% (58/347) [54] showed changes in the caloric test even up to 15 years after surgery. A small percentage of patients (2.6%) may experience vertigo lasting over 12 months, indicating permanent postoperative vestibular hypofunction [64] (Table 1).

One potential cause of late postoperative vertigo is the perilymphatic fistula, which occurs due to inadequate sealing around the prosthesis at the oval window. Its incidence is variable from one study to another, ranging from 1.3% to 10% [65,66,67,68]. Pedersen et al. [65] suggested that the cause may be inadequate sealing around the prosthesis in the oval window. A systematic review of the results and complications of stapes surgery confirms that perilymphatic fistula is a rare complication of stapes surgery and represents approximately one third of surgical revision cases [69]. Although its existence has been highlighted intraoperatively in only a few cases, the correlation between filling with tissue or fibrin glue and remission of symptoms suggests that the perilymphatic fistula is often underestimated [70].

The perilymphatic fistula can persist postoperatively if the hole around the prosthesis has not been closed, or may appear later if the graft or prosthesis moves as a consequence of increased pressure due to coughing or sneezing. Usually, complaints involve fluctuating hearing loss and vertigo, and the audiogram indicates deterioration of cochlear function [71]. Incidence can be reduced in the case of stapedectomy by placing a graft over the oval window [72].

Due to the risk of meningitis and hearing loss, the presence of perilymphatic fistula represents a serious complication. If a perilymphatic fistula is suspected and the symptoms do not improve with treatment, surgical exploration of the ear is necessary in order to close the fistula with a soft tissue graft. Persistence of the fistula can lead to irreversible hearing loss and persistence of vertigo [2].

According to Nakashima et al., the incidence of perilymphatic fistula can reach 22% in patients in whom the obliteration was performed with gelfoam and 4% in those in whom the obliteration was performed with tissue [73]. An older study comparing gelfoam, fat tissue, and fascia showed an incidence of perilymphatic fistula of 3.5% in case of gelfoam, 1.9% in case of fat tissue, and 0.6% in the case of fascia [74]. According to Lim et al. [75], from an auditory point of view fatty tissue is to be preferred in stapedotomy and fascia in stapedectomies.

Other causes of postoperative vertigo include irritation produced by a protracted prosthesis or a displaced one. Symptoms intensify when moving the head or during the Valsalva manoeuvre. The patient may experience dizziness related to hiccupping, burping, yawning, popping of the ears, and specific acoustic stimuli [76].

Reparative granuloma, which is a pyogenic inflammatory reaction, autoimmune or allergic reaction, or exuberant healing response, can occur in approximately 0.1% to 18% of cases after stapedectomy or stapedotomy [77,78,79,80]. Reparative granulomas occur more frequently after stapedectomy, and are characterized by sensorineural hearing loss along with vertigo. Typically, reparative granuloma manifests 7 to 15 days after surgery [81].

Persistent late vertigo can be due to bone fragments entering the vestibule during surgery; additional causes include direct compression of the saccule due to adhesion between the prosthesis and the tympanic membrane, Eustachian tube dysfunction, and Tullio phenomenon [82,83,84]. Stapes surgery can damage the inner ear and eventually leads to endolymphatic hydrops without decreasing the hearing threshold at low frequencies [85]. Endolymphatic hydrops (EH) can be associated with otosclerosis as a secondary condition following stapes surgery, when EH can occur immediately after the surgery or at a later time. Clinical manifestations include low-frequency fluctuating sensorineural hearing loss, episodic vertigo, tinnitus, and aural fullness. However, according to Halpin et al. [85] these symptoms are much more rare compared to the presence of histopathological findings on TB specimens of patients who underwent otosclerosis surgery.

Rarely, late-onset vertigo can be associated with pneumolabyrinth or barotrauma, a condition presented by Mandala et al. in a patient who started to have vertigo years after surgery [86]. Additionally, Gomes et al. [87] published a case report about a patient who came back 4 weeks after stapedectomy for displacement of the prosthesis and the graft. In most cases, this complication occurred a few weeks or months after the surgery. The diagnosis is based on HRCT scan showing the presence of air bubbles in the vestibule in patients with vertigo and a positive fistula test.

Several factors can contribute to prolonged vertigo, including age, sex, stapes surgery in the opposite ear, the seal around the prosthesis in the footplate, and postoperative hearing outcomes. A history of stapes surgery in the opposite ear has been identified as a significant predictive factor for prolonged nystagmus and subjective vestibular symptoms [55].

Intractable vertigo may be an indicator for revision surgery in otosclerosis cases [54]. Prompt diagnosis and appropriate management are crucial in addressing the underlying causes of late postoperative vertigo and improving patient outcomes.

## 7. Treatment of Vertigo

The treatment of vertigo in patients with otosclerosis depends on the underlying cause and the severity of symptoms.

The immediate postoperative vertigo during the surgical procedure and in the first few days resolves mostly with bed rest and symptomatic treatment within approximately 5 to 7 days, rarely lasting more than 4 weeks.

If vertigo symptoms persist or are caused by specific complications such as a perilymphatic fistula or malposition of the prosthesis, surgical intervention can be considered to obliterate a possible perilymphatic fistula, reposition the prosthesis, or take other corrective measures to address the underlying cause of vertigo.

Postoperative or late vertigo can be improved by treatment with the latest generation of bisphosphonates or by physical therapy, including vestibular rehabilitation exercises. These exercises aim to improve balance, reduce dizziness, and promote central compensation of the vestibular system.

Medical treatment of otosclerosis includes drugs that can directly influence bone metabolism, anti-inflammatory agents that address the inflammatory etiology, targeted (biological) therapies, and, last but not least, anti-measles vaccination [88].

Bone metabolism inhibitors aim to preserve hearing thresholds and alleviate symptoms such as tinnitus and vertigo associated with otosclerosis. Sodium fluoride, often combined with calcium carbonate and vitamin D, is used to slow down the progression of otosclerosis. It has been shown to reduce the deterioration of hearing loss and help to control tinnitus and vestibular symptoms. However, there are differing opinions as to its overall effectiveness in treating otosclerosis. Studies have stated that sodium fluoride can slow down the evolution of the disease in more than 50% of cases [89], while others have shown reduced efficiency in the treatment of otosclerosis [88].

Studies have shown that bisphosphonates can influence vestibular symptoms in patients with otosclerosis before or after surgery. Brookler and Tanyeri have reported that 54% of patients presented an improvement in vestibular symptoms after treatment with bisphosphonates, while 39% reported disappearance of dizziness and 35% presented improvement in the results of tests performed with a rotatory chair [90]. The newer generations of bisphosphonates (e.g., risedronate, zoledronate) have more favorable tolerability and are more powerful bone resorption inhibitors [91].

Bioflavonoids can reduce bone resorption by inhibiting the phosphodiesterase enzyme. While they might not significantly affect hearing loss, they have been found to significantly reduce tinnitus [92].

Vitamin D’s anti-inflammatory effects and vitamin A’s ability to inhibit osteoclast differentiation could potentially have a beneficial impact on otosclerosis [88,93].

Regarding anti-inflammatory agents, corticosteroids are often used in inner ear diseases, including otosclerosis. Transtympanic administration can be a solution to minimize systemic side effects. Nonsteroidal anti-inflammatory drugs have been considered for their inhibitory effect on bone resorption; however, there is a lack of long-term clinical data in otosclerosis [88,92].

From the class of immunosuppressive agents, only cyclosporine A has been studied, and there is limited data on the use of other immunosuppressive drugs in treating otosclerosis [94].

Emerging treatments such as anti-TNF-α agents (e.g., infliximab), recombinant human OPG (rhOPG), and other anti-osteoporotic targeted therapies (e.g., denosumab, odanacatib) hold potential for otosclerosis treatment, however, more long-term studies are needed [92].

Fluoride-based medications and bisphosphonates are among the treatments considered for medical management of otosclerosis-related hearing loss and associated symptoms such as tinnitus and vertigo.

## 8. Limitations of the Study

The reported percentages of balance disorders in otosclerosis patients both pre- and postoperatively varies significantly in the literature due to differences in study design and patient assessments. The main complaint in otosclerosis is typically hearing loss, with balance disorders often being secondary, which can lead to variations in the assessment and reporting of vestibular symptoms. Postoperative vertigo can vary depending on factors such as the individual patient, surgeon’s experience, surgical technique, and type of prosthesis used.

Advancements in technology have provided new tools for assessing vestibular deficits, allowing for more objective evaluation of vertigo. Objective measures can provide valuable information about vestibular function before and after surgery.

Multicentre studies with larger patient populations and standardized evaluation protocols would be beneficial in providing more comprehensive and reliable data on balance disorders in otosclerosis patients before and after surgery.

## 9. Conclusions

Vertigo is a common manifestation in otosclerosis both before and after surgical treatment. It can present as benign paroxysmal positional vertigo, vertigo attacks, or hydrops, as well as dizziness or light-headedness. While vertigo quite frequently appears immediately after surgery, the symptoms typically subside quickly with medical treatment, and only persist for longer periods of time in a very few cases.

When vertigo occurs months or years after the surgery, this can be an alarming signal of a complication that may require surgical reintervention. In such cases, careful evaluation and appropriate management are crucial in order to address the underlying cause and alleviate the vertigo symptoms.

Overall, understanding the occurrence and characteristics of vertigo in otosclerosis both pre- and postoperatively is essential for effective diagnosis, treatment, and patient management. Further research and standardized assessment protocols are needed in order to provide more comprehensive data and improve the management of balance disorders in otosclerosis patients.

## Figures and Tables

**Table 1 medicina-59-01485-t001:** Summarised data on vertigo associated with otosclerosis.

	Preoperative	Intraoperative	Immediate Postoperative	Late Postoperative
Incidence	8.6–30%	2.1%	3.4–70%	0.5–17%
Causes	-endolymphatic hydrops -detachment of the otoconia–cupular deposits -vestibular endorgan and/or neural degeneration -hydrolytic enzymes, cytokines	-manipulation of the footplate -suctioning in the middle ear/oval window -floating footplate	-serous and chemical labyrinthitis -suction of perilymph -penetration of instrument or prosthesis into the vestibule -traumatization of the utricle or saccule -release of proteolytic enzymes -antigen-antibody reaction -changes in labyrinthine fluids pressure -reduction of blood supply to the labyrinth -stapes surgery in the opposite ear	-perilymphatic fistula -secondary endolymphatic hydrops -irritation by prolonged prosthesis -reparative granuloma -bone fragments entering the vestibule -pneumolabyrinth, barotrauma -stapes surgery in the opposite ear
Progress	-Fluctuating -Permanent	Usually temporary	Usually temporary	-Temporary -Fluctuating -Permanent
Treatment	-Sodium fluoride -Bisphosphonates	-Vertiginous drugs -Bed rest	-Vertiginous drugs -Bed rest	-Sodium fluoride -Bisphosphonates -Vestibular rehabilitation exercises

## Data Availability

Data sharing not applicable.

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
