# Peer review of "Vertigo Associated with Otosclerosis and Stapes Surgery—A Narrative Review"

_medicina, 2023, doi:10.3390/medicina59081485_

Round 1
Reviewer 1 Report
Please see the attached file.

Reviewer 2 Report
This is a very remarkable and well structured paper. It should be informative and of great help for surgeons facing vestibular disorders in otosclerosis long-time denied.
By going to the different points some minor comments.
1. OK
2. Histological presentation: superb. But no mention to molecular basis.
3. OK
4. Preoperative vertigo. This is a good explanation of causes and state of the art in vestibular exam. However it misses (and my suggestion is to go deep on that ) that patients with otosclerosis could have any other type of vertigo (spontaneous or positional) and even non-vertigo symptoms but chronic dizziness: PMID: 21488576. The causes are set but is not mentioned.
Also it is interesting to include whether or not based on pre-operative vertigo if is there any contraindication for surgery.
Last. The need for end-lymphatic hydrops assessment pre-operative (10.1080/00016489.2016.1232862). In which cases?
5. Looks like there is no mention to semicircular canal damage...is that so?
6. Reparative granuloma: please mention the type of hearing loss; also given this to be a sub-acute disorder the term progressive looks very unclear.
No mention to secondary hydrops. Fluctutatinh hearing loss just due to fistula is not very much frequent.
7. This is a short chapter of a great importance. Given the wide variety of alternative treatments for vertigo and the specific for otosclerosis (DOI: 10.1007/s00405-012-2126-0) this needs some re-rowrk if the authors want to be more practical fro the audience.
